# Comparison of triple-DMEK to pseudophakic-DMEK: A cohort study of 95 eyes

**Axelle Semler-Collery**[1], **Florian Bloch**[1], **George Hayek**[1], **Christophe Goetz**[2], **Jean Marc Perone**[1,3]*

**1** Ophthalmology Department, Mercy Hospital, Metz-Thionville Regional Hospital Center, Metz, France, **2** Research Support Unit, Mercy Hospital, Metz-Thionville Regional Hospital Center, Metz, France, **3** Institut Jean Lamour, Team 404, UMR 7198 Nanomaterials and Health, Lorraine University, Nancy, France

* jm.perone@chr-metz-thionville.fr

**Data Availability Statement:** The datasets generated during and/or analyzed during the current study are not publicly available according to French Law No. 2018-493 of June 20, 2018 on the protection of personal data (The General Data

## Abstract

Previous comparative studies show that triple Descemet membrane endothelial keratoplasty (DMEK) (i.e. phacoemulsification followed immediately by DMEK) has either equivalent or better visual outcomes than DMEK in pseudophakic patients. To resolve this discrepancy, a retrospective cohort study was conducted. All consecutive patients with Fuchs Endothelial Corneal Dystrophy who underwent triple or pseudophakic DMEK in 2015–2019 in a tertiary-care hospital (France) and were followed for >12 months were compared in terms of best spectacle-corrected visual acuity (BSCVA), final refractive outcomes, and endothelial-cell loss at 12 months as well as rebubbling rates. The triple-DMEK (40 eyes, 34 patients) and pseudophakic-DMEK (55 eyes, 43 patients) groups were similar in terms of age and other baseline variables. They also did not differ in final BSCVA (both 0.03 logMAR), final endothelial-cell loss (54% vs. 48%), or astigmatism (-1.25 vs. -1 D). At 12 months, triple-DMEK associated with significantly smaller residual hyperopia (0.75 vs. 1 D; p = 0.04) and spherical equivalence (0 vs. 0.5 D; p = 0.02). Triple-DMEK also tended to associate with more frequent rebubbling (40% vs. 24%, p = 0.09). In conclusion, while triple-DMEK and pseudophakic-DMEK achieved similar visual acuity improvement, triple-DMEK was superior in terms of final sphere and spherical refraction but also tended to have higher complication rates.

## Introduction

The management of corneal endothelial deficits secondary to Fuchs endothelial corneal dystrophy (FECD) and pseudophakic bullous keratopathy underwent a revolution 15 years ago when Melles [1–4] (with significant contributions from others [5–10]) developed two posterior lamellar corneal transplantation procedures, namely, Descemet stripping automated endothelial keratoplasty (DSAEK) and Descemet membrane endothelial keratoplasty (DMEK). Although DMEK is more difficult to perform than DSAEK, it is often the treatment of choice for FECD [1, 11, 12] because it associates with better recovery, postoperative best spectacle-corrected visual acuity (BSCVA) [12–14], contrast [15, 16], immune rejection [17, 18], patient satisfaction [12, 19, 20], and final posterior residual corneal higher-order aberrations [21].

Protection Regulation (Regulation (EU) 2016/679) (GDPR: article 9) but are available from the Clinical Research Support Platform (Plateforme d'Appui à la Recherche Clinique [PARC]) of the Regional Central Hosital (CHR) of Metz-Thionville on reasonable request (email: projet-recherche@chrmetz-thionville.fr, tel: +33 3 87 17 98 82). All non-archived data is subject to daily backups while all archived data is subject to duplicate storage at two different sites. This data processing is compliant with a baseline reference methodology (MR-004) for which the CHR Metz-Thionville signed a compliance commitment on October 8, 2018.

**Funding:** The author(s) received no specific funding for this work.

**Competing interests:** The authors have declared that no competing interests exist.

FECD was first described by Ernst Fuchs in 1910 [22] and presents as loss of corneal transparency due to corneal edema. This edema is the result of the gradual loss of corneal endothelial cells, which can no longer adequately pump fluid out of the corneal stroma [23]. FECD is the most common corneal dystrophy: in 2016, it accounted for 23% of all corneal transplants in the USA [24]. Caucasians and women are particularly prone to FECD: a cohort study in Reykjavik, Iceland reported that 11% and 7% of all women and men had primary corneal guttae, respectively [25]. The disease is age-related, mostly manifesting after the fourth decade [23]. Due to this, patients with FECD often present with other age-related ocular comorbidities, including cataract [26], whose prevalence in Europeans increases from 5% in 52–62-year-olds to 64% in >70-year-olds [27]. Consequently, the advent of DMEK for FECD quickly triggered a debate about whether phacoemulsification and DMEK should be conducted in stages or in the same operation (known as phaco-DMEK or triple-DMEK). Three retrospective comparative studies on largely FECD patients have addressed this question in the last decade, with conflicting results: whereas two showed that triple-DMEK and DMEK in pseudophakic eyes yielded equivalent postoperative BSCVA [28, 29], the third showed that triple-DMEK associated with better postoperative BSCVA [30]. However, the two groups in two studies differed markedly in terms of age and/or preoperative BSCVA and one study had a small sample size (S1 Table).

Therefore, to address this issue further, we conducted a retrospective cohort study comparing triple-DMEK to pseudophakic-DMEK in terms of BSCVA and other outcomes at 12 months.

## Materials and methods

### Ethics

This retrospective cohort study was conducted in accordance with the principles of the Helsinki Declaration. The study was approved by the ethics committee of the French Society of Ophthalmology (IRB 00008855 Société Française d'Ophtalmologie IRB#1) and was registered in clinicaltrials.gov (No. NCT03355924). All patients provided informed consent for surgical management. All patients were informed before surgery that their surgery-related data might be used for research purposes. All consented to this possibility. The consent procedure was conducted in accordance with the reference methodology MR-004 of the National Commission for Information Technology and Liberties of France (No. 588909 v1).

### Patient selection

The patient cohort consisted of all consecutive adult (≥18 years) patients with FECD and BSCVA ≤0.3 logMAR who underwent triple-DMEK or pseudophakic-DMEK in January 2015–January 2019 in the Department of Ophthalmology of the Metz-Thionville Regional Hospital Center (Metz, France) and who were followed up for at least 12 months. Patients were excluded if the patient was lost to 12-month follow-up and/or the patient had eye damage that could influence visual acuity recovery (age-related macular degeneration, advanced diabetic retinopathy, advanced glaucoma, history of uveitis, history of eye surgery other than cataract surgery, or deep amblyopia).

### Preoperative assessments

All patients underwent a complete clinical examination before surgery. Distance visual acuity was determined by using Monoyer's visual acuity scale; the scores were then converted to logMAR. Near visual acuity was determined by using the Parinaud scale. Intraocular pressure was

measured. Optical coherence tomography (OCT) was used to examine the macula (RS-3000 OCT RetinaScan advance, NIDEK Co., Ltd., Japan). Anterior-segment OCT was also conducted to measure corneal thickness. Specular microscopy (CEM-530 NIDEK, Japan) was performed to assess endothelial-cell density (ECD).

## Surgical techniques

In all cases, lower iridotomy with the Nd-YAG laser (Laser ex-Super Q; Ellex Europe, Medical Quantel, Cournon-d'Auvergne, France) was conducted at the preoperative consultation 1–2 months before surgery.

All surgeries were performed by the same experienced surgeon (JMP). In all cases except two, triple-DMEK and pseudophakic-DMEK procedures were performed under general anesthesia. The exceptions (one triple-DMEK, one pseudophakic-DMEK) were performed under locoregional anesthesia due to general anesthesia contraindications. All grafts were from two French regional tissue banks (Besançon or Nancy). All were preserved by organo-culture and had a requested ECD greater than 2400 cells/mm$^2$.

All DMEK surgeries were conducted as described previously [2]. Thus, the corneal graft was prepared by 8 mm-diameter trepanation with a Hanna's microtrepan (Busin Punch 17200D 8mm single use; Moria SA, Antony, France) and the Descemet membrane with the endothelium was dissected with a disposable curved monofilament forceps (Single Use Tying Forceps Curved 5mm Platform 17501, Moria SA, Antony, France). The graft was then stained with VisionBlue dye (VisionBlue, 0.5-ml syringe; D.O.R.C. Dutch Ophthalmic Research Center, Zuidland, Netherlands), marked on its stromal side with a capital E or F for later orientation, and inserted into the D.O.R.C injectable system (30G Curved cannula for air injection; D.O.R.C. Dutch Ophthalmic Research Center, Zuidland, Netherlands). The patient's cornea was then prepared: the main paracentesis was conducted at a supero-temporal (for the right eye) or supero-nasal (for the left eye) location with a Worst 2.2-mm blade (Securityblade BD, Xstar 2.2-mm, 45 degrees, 37822; Beaver-Visitec International, USA). The secondary incision was then made with a Worst 15 blade (ophthalmic knife 15 degrees; ALCON, Ruel Malmaison, France). A sterile air infusion was established, and a 9-mm diameter circle of Descemet membrane was dissected by using an inverted Sinskey Price hook (Single Use Price Reverse Hook Sim 17302; Moria SA, Antony, France). The tissue was then ablated by using the same Sinskey Price hook and an inverted spatula (90th single use Spatula 17303; Moria SA, Antony, France) and the graft was installed. First, the temporal incision was slightly enlarged to >4 mm by using the Worst 2.2mm knife (SecurityKnife BD, Xstar 2.2-mm, 45 degrees, 37822; Beaver-Visitec International, USA). The graft was then injected into the anterior chamber by using the D.O.R.C. injectable system (30G Curved cannula for air injection; D.O.R.C. Dutch Ophthalmic Research Center, Zuidland, Netherlands). The graft was carefully moved and centered by external manipulation. A sterile air bubble was then injected under the graft to place it at the posterior side of the cornea. A corneal suture was placed with 10.0 Nylon and the knot was secondarily buried.

In triple-DMEK, phacoemulsification was initially performed according to a standard subluxation method [31] using an easy to remove ophthalmic viscosurgical device (Duovisc, Alcon, Rueil Malmaison, France). At the end of the procedure, the device was then completely suctioned off, MIOSTAT$^®$ 0.01% (Carbachol, ALCON, Rueil Malmaison, France) was injected, and the incision edges was hydrosutured. This procedure was then followed by DMEK, which was conducted exactly as described above. Note that the refractive target in triple-DMEK was a residual myopia of about -0.5 to -1.00 diopters to compensate for the hypermetropizing effect of the surgery.

## Postoperative care

All patients were followed closely postoperatively by consultations on days 1, 8, and 15, months 1, 3, 6, and 12, and then annually. All patients received early postoperative treatment, namely, antibiotic eye drops coupled with a corticosteroid (MAXIDROL® Dexamethasone + Neomycin Polymyxine B; ALCON, Rueil Malmaison, France) 4 times a day for the first 3 months and an ophthalmic ointment with vitamin A (Vitamin A dulcis; ALLERGAN, Courbevoie, France) 3 times a day for 3 months. After 3 months, this treatment was replaced with long-term low dose corticosteroid eye drops (FLUCON®; NOVARTIS Pharma, Rueil-Malmaison, France) twice a day for about one year.

The same ophthalmic assessments that were conducted before surgery were repeated at each follow-up consultation. Postoperative complications such as cystoid macular edema (CME) or graft detachment were documented during these consultations. Patients who developed CME and experienced a drop in visual acuity were treated with oral acetazolamide (Diamox®; Sanofi, France; one 250 mg tablet 3 times/day for one month) and NSAID (indomethacin 0.1%; Chauvin, France; 4 times/day for one month). If central graft detachment was observed or more than a third of the graft had detached, rebubbling with an air tamponade was performed under topical anesthesia by puncturing the anterior chamber and then injecting air into it. If rebubbling sessions (maximum four) did not result in graft adherence or the cornea did not thin after 3 months of follow-up, the transplant was considered to have failed and a second transplant was scheduled.

## Primary and secondary study outcomes

The following data were collected retrospectively from the prospectively maintained medical charts: patient age and sex; total operative time; graft donor age and ECD; the time between phacoemulsification and DMEK for the pseudophakic-DMEK patients; preoperative and postoperative (day 8 and 15 and months 3, 6, and 12) BSCVA; preoperative and 12-month refractive data; postoperative (months 3, 6, and 12) ECD; preoperative and 12-month central corneal thickness; 12-month corneal thinning; and postoperative complications, namely, CME, graft detachment that required anterior chamber rebubbling, repeated rebubbling, and graft failure.

The primary study outcome was 12-month BSCVA. The secondary study outcomes were BSCVA at earlier postoperative timepoints, endothelial cell loss (ECL) at 3, 6, and 12 months, 12-month central corneal thickness, 12-month refractive outcomes, and complication rates.

## Statistical analyses

The data were complete for all patients included in the analysis. Patients who had transplant failure (defined as persistent corneal edema 3 months after transplantation) were included in

**Table 3. Refractive outcomes of the study groups at 12 postoperative months.**

| Variable | Triple-DMEK 40 eyes (34 patients) | Pseudophakic-DMEK 55 eyes (43 patients) | P value* |
|---|---|---|---|
| Sphere, D | 0.75 (-1.75–5.75) | 1 (-1.0–4.50) | **0.04** |
| Cylinder, D | -1.25 (-4.25–0) | -1 (-3.25–0) | 0.37 |
| Axis | 90 (0–170) | 95 (0–180) | 0.22 |
| SE, D | 0 (-1.88–4.88) | 0.5 (-1.5–4.25) | **0.02** |

The data are expressed as mean±standard deviation (range).

* The groups were compared by Wilcoxon test.

DMEK, Descemet membrane endothelial keratoplasty; SE, Spherical equivalence.

the cohort but excluded from statistical analysis. The continuous data were expressed as mean ±standard deviation while the categorical variables were expressed as *n* (%). The triple-DMEK and pseudophakic-DMEK groups were compared by using Student's *t*-test for quantitative outcomes with normal distribution, Wilcoxon test for quantitative outcomes with non-normal distribution, and Fisher's exact test for qualitative outcomes. P values <0.05 were considered to indicate statistical significance. All statistical analyses were conducted by using SAS software (version 9.4, SAS Inst., Cary, NC, USA).

## Results

### Patient disposition in the study

In total, 123 eyes (105 patients) underwent DMEK during the 2015–2019 study period. Of these, 48 and 75 eyes underwent triple-DMEK and pseudophakic-DMEK, respectively. Three triple-DMEK eyes (6%) were excluded because they had a non-FECD pathology. Twelve pseudophakic-DMEK eyes (16%) were excluded because they had bullous pseudophakic keratopathy (*n* = 4), another non-FECD pathology (*n* = 6), or were lost to follow-up (*n* = 2). Consequently, 108 eyes were included in the study. Of these, 45 eyes and 63 eyes received triple-DMEK and pseudophakic-DMEK, respectively. During the study, five (11%) and eight (13%) of these respective eyes required new transplants and were excluded from the statistical analysis. Thus, in total, 40 triple-DMEK eyes and 55 pseudophakic-DMEK eyes were included in the statistical analyses (Fig 1).

### Demographic and baseline clinical characteristics of the study groups

In the pseudophakic-DMEK group, the median (range) duration between phacoemulsification surgery and DMEK surgery was 32 (1–264) months. The triple-DMEK and pseudophakic-DMEK groups were similar in age, sex, preoperative BSCVA, preoperative refractive variables, and preoperative central corneal thickness. The two groups were also similar in terms of graft donor age and graft ECD. The triple-DMEK group had a significantly longer total operative time (p<0.0001), which reflected the fact that two procedures were conducted instead of one (Table 1).

### Primary and secondary study outcomes

The triple-DMEK and pseudophakic-DMEK groups did not differ significantly in terms of BSCVA at any of the postoperative timepoints: at 12 months, the mean BSCVAs were 0.03 and 0.03 logMAR, respectively (Fig 2). Triple-DMEK associated with slightly but consistently lower ECD but these differences did not achieve statistical significance. At 12 months, the mean ECL was 54% and 48% (p = 0.18), respectively (Table 2). However, compared to the pseudophakic-DMEK group, the triple-DMEK group had a significantly smaller residual hyperopia, namely, 0.75 (range, -1.75–5.75) D *vs*. 1.0 (-1.0–4.5) D (p = 0.04). Their postoperative spherical equivalent was also smaller, namely, 0 (-1.88–4.88) D *vs*. 0.5 (-1.5–4.25) D (p = 0.02). The groups did not differ in astigmatism (Table 3). Central corneal thickness in the two groups evolved in the same way (Fig 3). Corneal thinning at 12 months was similar (86 *vs*. 106 μm; p = 0.11).

In terms of complications, the triple-DMEK and pseudophakic-DMEK groups required five (11%) and eight (13%) second transplants with DMEK or DSAEK (p = 0.77). All second transplants were successful. Rebubbling tended to be more frequent in the triple-DMEK group: 40% required one re-bubbling *versus* 24% for the pseudophakic-DMEK group (p = 0.09). The two groups did not differ in second rebubbling rates (5% *vs*. 7%, p = 0.69)

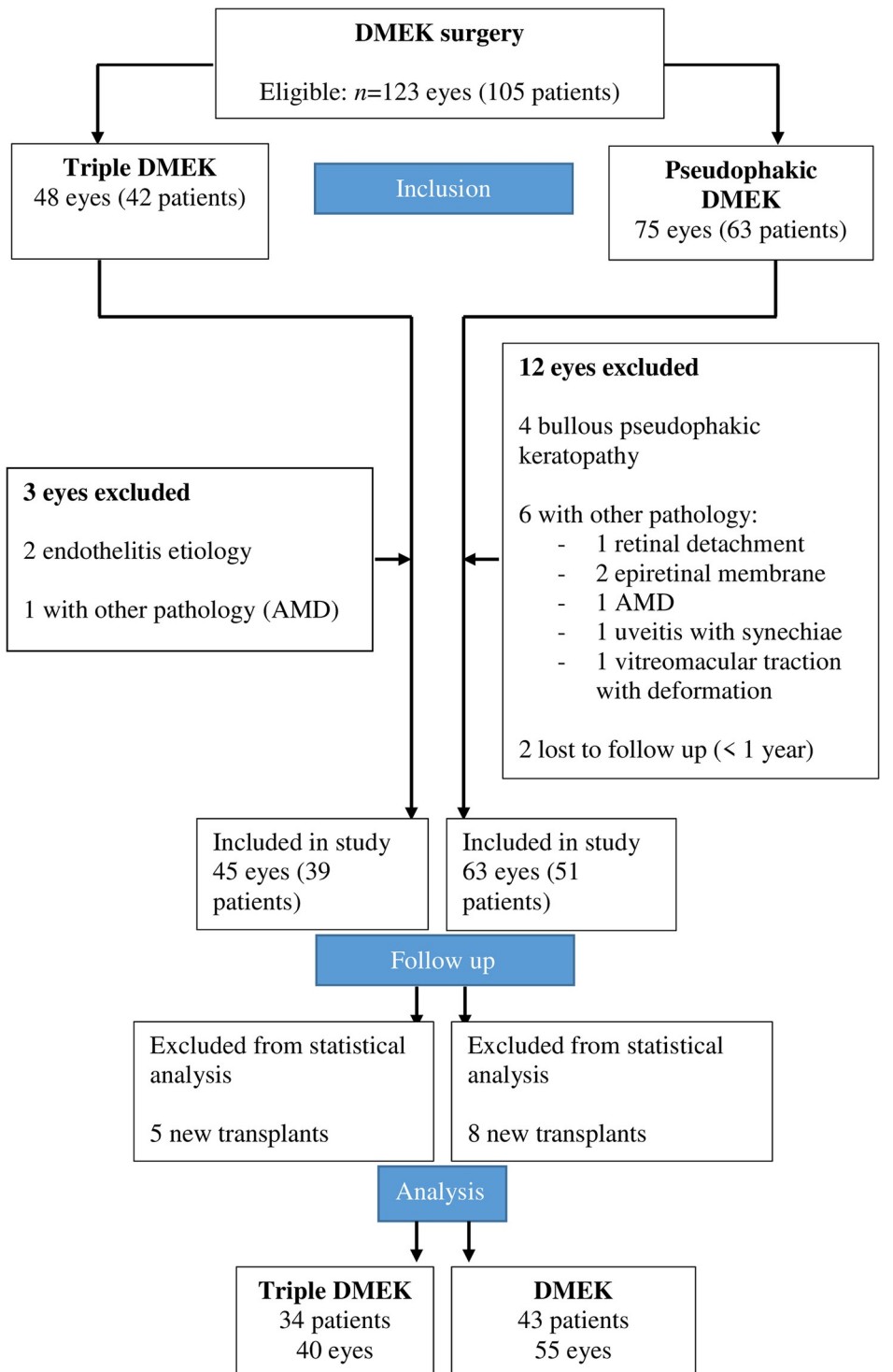

**Fig 1. Flow chart depicting the disposition of the subjects during the study.** AMD, age-related macular degeneration; DMEK, Descemet membrane endothelial keratoplasty.

**Table 1. Baseline and operative characteristics of the study groups.**

| Variable | Triple-DMEK 40 eyes (34 patients) | Pseudophakic-DMEK 55 eyes (43 patients) | P value* |
|---|---|---|---|
| Patient age, y | 70 (52–84) | 72 (54–86) | 0.10 |
| Female sex | 27 (79) | 37 (86) | 0.98 |
| Preop. BSCVA, logMAR | 0.50 (1.7–0.3) | 0.50 (1.7–0.3) | 0.13 |
| Preop. Sphere, D | 1 (-8–4.75) | 0.75 (-2.5–3.75) | 0.50 |
| Preop. Cylinder, D | -1 (-5.25–0) | -1.25 (-2.5–0) | 0.32 |
| Preop. Axis, degrees | 82.5 (0–164) | 90 (0–177) | 0.39 |
| Preop. Sph.Eq, D | 0.63 (-8.6–4) | 0.06 (-2.75–2.625) | 0.31 |
| Preop. CCT, μm | 611 (530–775) | 620 (548–786) | 0.41 |
| Graft donor age, y | 77 (49–93) | 76 (30–99) | 0.97 |
| Graft ECD, cells/mm$^2$ | 2545 (2230–2950) | 2540 (2250–2900) | 0.13 |
| Total operative time, min | 35 (30–60) | 30 (20–60) | **<0.0001** |

The data are expressed as median (range) or *n* (%).

* The groups were compared by Student's *t*-test, Wilcoxon test or Fisher's exact test, as appropriate.

BSCVA, best spectacle corrected visual acuity; CCT, central corneal thickness; DMEK, Descemet membrane endothelial keratoplasty; ECD, endothelial cell density; Preop., preoperative; Sph.Eq., spherical equivalent.

(Fig 4). CME occurred with equivalent frequency in the two groups (*n* = 2 and 3, respectively; both 5%, p = 0.92).

## Discussion

Our retrospective cohort study showed that compared to patients who underwent DMEK on average 47 months after phacoemulsification, triple-DMEK patients had similar 12-month

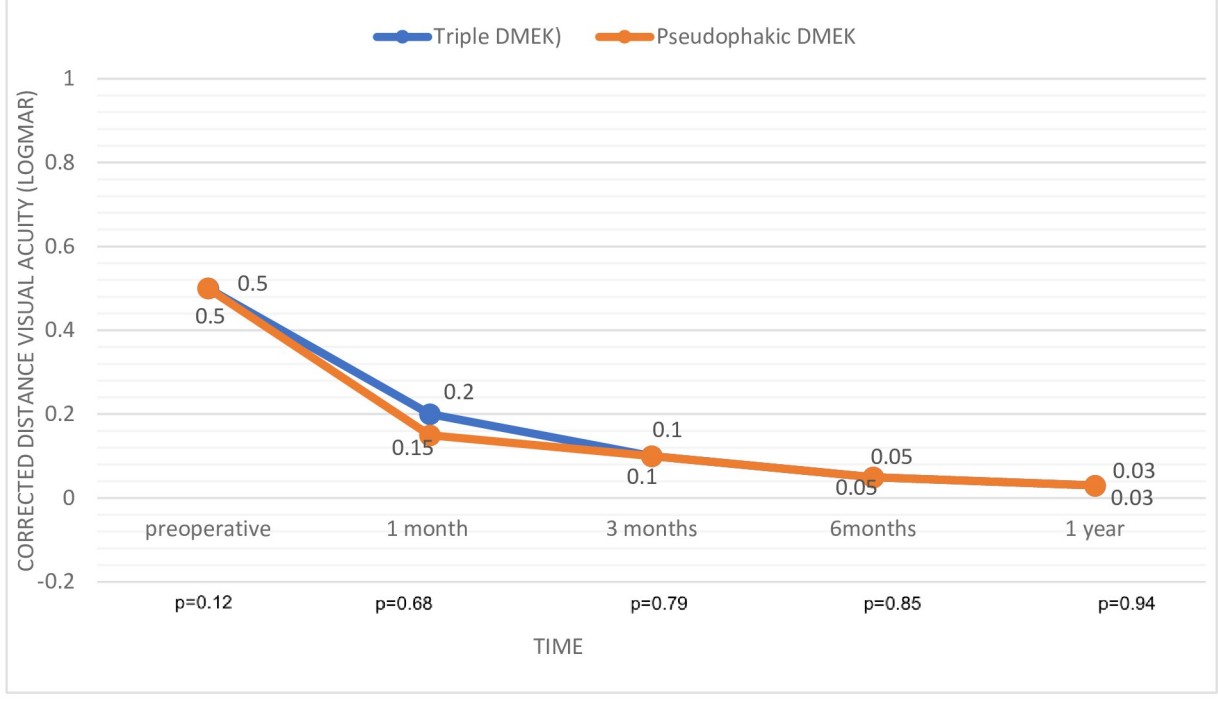

**Fig 2. Change in visual acuity during follow-up.** DMEK, Descemet membrane endothelial keratoplasty.

**Table 2. Endothelial cell density and loss outcomes of the study groups.**

| Timepoint | Triple-DMEK 40 eyes (34 patients) | Pseudophakic-DMEK 55 eyes (43 patients) | P value[*] |
|---|---|---|---|
| Endothelial cell density, cells/mm$^2$ | | | |
| Preoperative | 2510±200 (2000–2900) | 2570±196 (2080–2950) | 0.13 |
| 3 Months | 1430±470 (640–2600) | 1570±370 (634–2239) | 0.12 |
| 6 Months | 1330±430 (666–2400) | 1460±360 (535–2240) | 0.13 |
| 12 Months | 1180±480 (509–2831) | 1300±380 (494–2032) | 0.19 |
| Endothelial cell loss, % relative to baseline | | | |
| 3 Months | 44±18 | 39±15 | 0.18 |
| 6 Months | 48±16 | 43±15 | 0.17 |
| 12 Months | 54±17 | 48±15 | 0.18 |

The data are expressed as mean±standard deviation (range).

Endothelial cell loss = % loss relative to baseline.

[*] The groups were compared by Student's *t*-test.

DMEK, Descemet membrane endothelial keratoplasty.

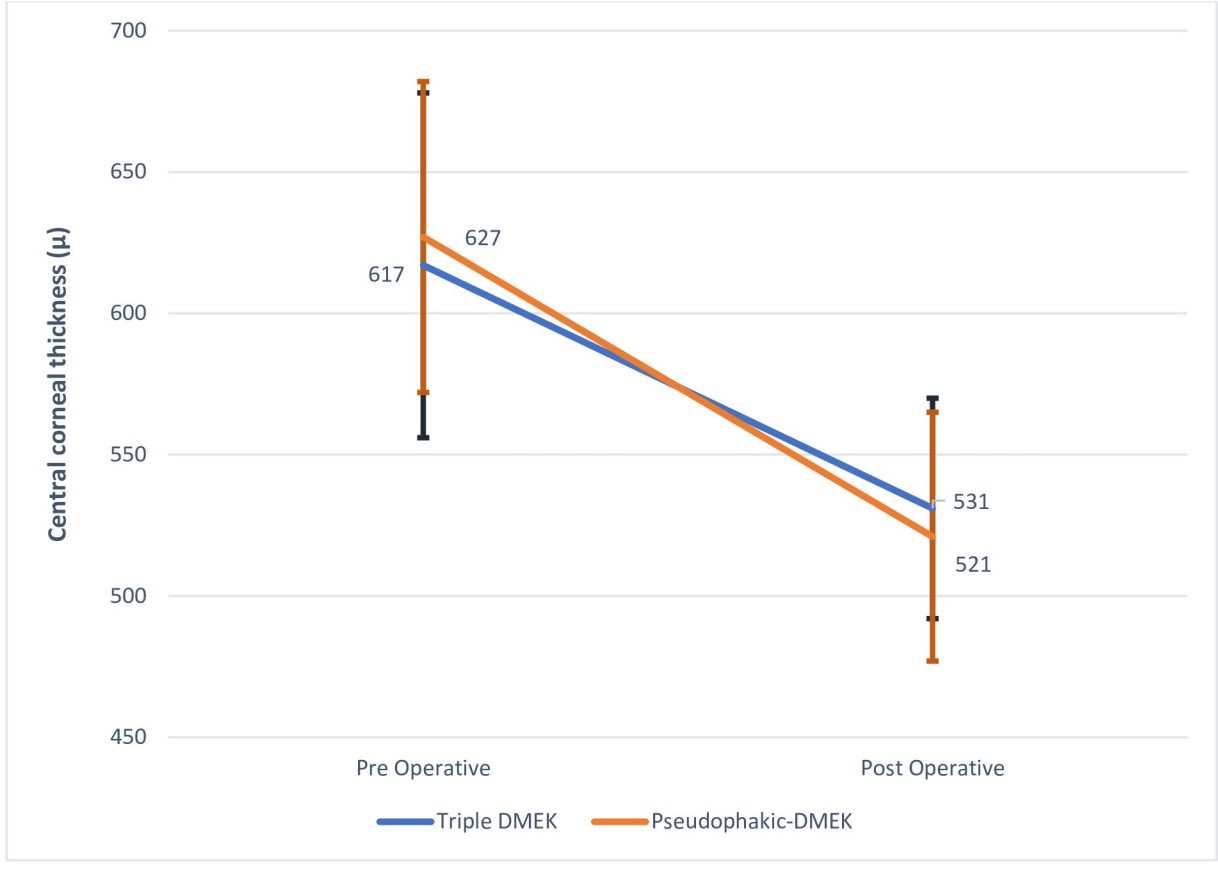

**Fig 3. Change in central corneal thickness in the study groups.** DMEK, Descemet membrane endothelial keratoplasty.

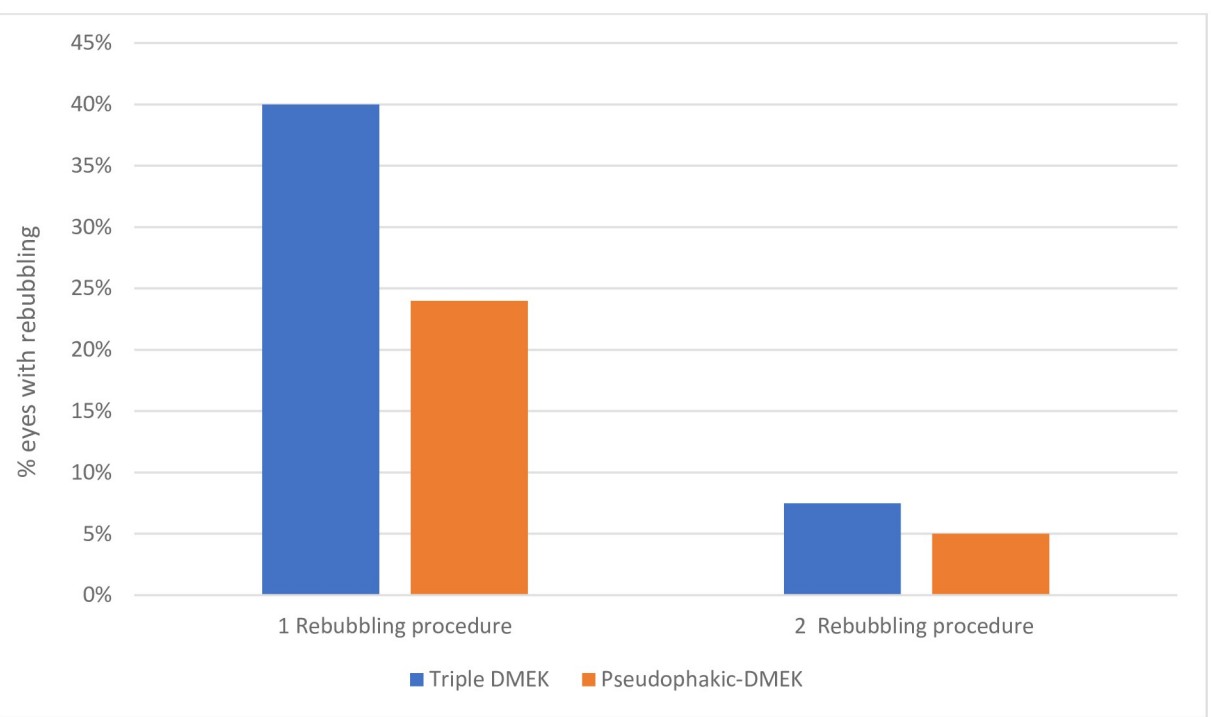

**Fig 4. Frequency of one or two rebubbling sessions after surgery.** DMEK, Descemet membrane endothelial keratoplasty.

visual acuity recovery, ECD and ECL, cylinder outcomes, central corneal thickness, and second transplantation and CME rates. However, the triple-DMEK group had significantly smaller residual hyperopia (p = 0.04) and spherical equivalent (p = 0.02) and a tendency to more frequent rebubbling (p = 0.09).

## Visual acuity

To date, three retrospective cohort studies have compared triple-DMEK and pseudophakic-DMEK in FECD. Our visual recovery results are consistent with two. Thus, Shahnazaryan *et al.* showed that BSCVA at 1 year was excellent in 80 triple-DMEK and 34 pseudophakic-DMEK patients [28]. A small study by Ighani *et al.* also did not observe visual acuity differences between triple-DMEK (*n* = 16) and pseudophakic-DMEK (*n* = 8) at a mean follow-up duration of 13 months (range, 3–26) months [29]. These findings are also consistent with an earlier case-series study by Laaser *et al.* on 61 triple DMEK cases in FECD, which showed comparable visual outcomes to DMEK alone in the literature [32]. By contrast, the large study of Chaurasia *et al.* found that triple-DMEK (*n* = 200) associated with significantly better median BSCVA than pseudophakic-DMEK (*n* = 292) at 6 months (20/20 *vs.* 20/25; p<0.0001) [30]. The discrepancies between these studies may reflect baseline differences between the two DMEK groups, which were present in all three comparative studies. First, the triple-DMEK patients were significantly younger than the pseudophakic-DMEK patients in the studies by both Shahnazaryan *et al.* [28] and Chaurasia *et al.* [30] (both p<0.0001). Second, compared to the pseudophakic-DMEK patients, the triple-DMEK patients in the Chaurasia *et al.* [30] and Ighani *et al.* [29] studies had better and worse preoperative BSCVA, respectively (p<0.0001 and p = 0.09). These baseline differences are important because Chaurasia *et al.* demonstrated with multivariate analysis that a younger age associated significantly with better postoperative

visual acuity [30]. Moreover, preoperative visual acuity is well-known to correlate positively with postoperative visual acuity [33]. Our study does not suffer from these baseline differences (age, p = 0.10; preoperative BSCVA, p = 0.48). Thus, our study shows that DMEK coupled with phacoemulsification yields satisfactory visual results. This has also been observed for DSAEK combined with phacoemulsification [34, 35].

## Endothelial cell loss

Our study showed that while triple-DMEK had slightly but consistently lower ECD than pseudophakic-DMEK at 3, 6, or 12 months, these differences did not achieve statistical significance: at 12 months, the average ECD was 1180 and 1300 cells/mm$^2$, respectively (p = 0.19). ECL at 6 months was 48% and 43%, respectively (p = 0.17). This is consistent with the relatively wide range reported by previous studies on 6-month ECL after DMEK (25–60%) [8, 9, 28, 30, 36, 37]. Moreover, Chaurasia et al. did not find that triple-DMEK differed from pseudophakic-DMEK in terms of 6-month ECL (both 26%) [30]. However, Shahnazaryan et al. showed that triple-DMEK associated with higher ECL (41% vs. 33%, p = 0.034) [28]. Again, this discrepancy may relate to baseline differences between the studies, which may have unpredictable effects on study outcomes. Nonetheless, the two groups in Shahnazaryan et al. had similar graft survival rates, which associates closely with ECL [38]; the two groups in our study also showed similar graft survival rates. It is unclear why the study of Shahnazaryan et al. differs from ours and Chaurasia et al. in ECL. While graft ECD and donor age associate with postoperative ECL [39, 40], the two groups did not differ in terms of these variables in our study and that of Shahnazaryan et al. and the Chaurasia et al. (the Ighani et al. paper did not compare the surgical groups in terms of donor variables) [28–30]. The discrepancy may therefore relate to other, more nebulous surgical factors such as more marked release of preoperative fibrin during triple-DMEK due to quick changes from mydriatic drugs to miotic drugs, different intervention durations, and contact between the endothelium of the graft and the new implant within the anterior chamber [28, 30].

## Rebubbling

We observed that triple-DMEK tended to associate with greater rebubbling rates than pseudophakic-DMEK (40% vs. 24%; p = 0.09). It is possible that with a larger sample size, this value would have become significant. The literature regarding this issue is discordant: some studies found that triple-DMEK associates with higher levels of postoperative graft detachment [41–43] whereas others did not observe that triple-DMEK associates with worse rebubbling rates [28, 30, 44, 45]. Again, the reason for these discrepancies is unclear but it may relate to differences between studies in terms of surgical factors. For example, filling the anterior chamber with <75% air is a known risk factor for graft detachment [42, 43]. Consequently, air is often replaced with SF6 gas, since it improves tamponade time without altering the new endothelium [34, 43, 46]. Since triple-DMEK involves the repeated introduction of instruments and greater incision permeability due to the implant, it may associate with faster loss of air postoperatively [43]. To prevent this, some surgeons apply sutures to close the main incision, although this has not yet been shown to reduce rebubbling rates [43]. The use of viscoelastic substances in triple-DMEK could also increase postoperative graft detachment rates [30, 41, 42]. In our study, we strictly used air in the anterior chamber, sutured the main incisions in both groups, and very carefully washed out the viscoelastic substances at the end of phacoemulsification. Nonetheless, it should be noted that one of the graft failure cases in the triple-DMEK group was due to the central persistence of small amounts of viscoelastic substance between the graft and the underlying stroma. Thus, residual viscoelastic substance could

possibly explain why we and others observed greater rebubbling rates in triple-DMEK compared to pseudophakic-DMEK. Further studies with larger sample sizes are needed to directly compare triple-DMEK and pseudophakic-DMEK in terms of rebubbling rates.

## Cystoid macular edema

Our rate of postoperative CME was 5% in both groups. Other studies have also reported that triple-DMEK does not associate with higher CME rates than pseudophakic DMEK, even though the rates reported tend to be higher than ours (7.5–13.8%) [40, 47, 48].

## Postoperative refractive outcomes

In our study, triple-DMEK tended to associate with less residual hyperopia (0.84 *vs*. 1.37 D; p = 0.06). It also associated with significantly less spherical equivalence (0.2 *vs*. 0.77 D; p = 0.03). To our knowledge, this is the first time these differences have been observed between triple-DMEK and pseudophakic-DMEK. In fact, very few studies seem to have compared these two surgical techniques in terms of final refractive outcomes. The only one we found was that by Ighani *et al*., who did not find differences in spherical equivalence [29]. However, we observed that 38% of the triple-DMEK patients (and 29% of pseudophakic DMEK patients) had less than half a spherical diopter, respectively, which is similar to that observed by Gundlach *et al*.: 35% of their triple-DMEK patients had less than half a spherical diopter [41]. The better spherical equivalence after triple-DMEK reflects the fact that by conducting phacoemulsification at the same time as DMEK, we can preventively correct the slight hypermetropization that is observed after DMEK (whereas we cannot do that in pseudophakic DMEK). Our findings suggest that while the residual postoperative hyperopia after triple-DMEK is good, we can compensate for it further by choosing the correct implant that will, as much as possible, yield final emmetropia. The good refractive results after triple-DMEK also make it possible to consider using toric implants for significant cylindrical abnormalities.

## Overall study conclusions

In our study, triple-DMEK and pseudophakic-DMEK both associated with marked improvements in postoperative visual acuity. Triple-DMEK associated with significantly better final spherical refractive outcomes, although it also tended to induce more rebubbling and ECL. Evaluation of the long-term consequences of triple-DMEK may be needed to ensure its relative safety. Nonetheless, our findings are important because combining phacoemulsification with DMEK prophylactically prevents progression of corneal impairment and eliminates a second surgery, thus reducing surgical risk [32]. It is also more cost-effective for the patient [32, 34, 46].

## Discussion of a recent third approach (DMEK in phakic patients)

Notably, there is recent debate in the field that centers on a third variation of phacoemulsification and DMEK surgery, namely, where DMEK is conducted first and is then followed by phacoemulsification at a later time point. This variation reflects the very satisfactory visual results after DMEK and the growing realization that performing DMEK before phacoemulsification could ultimately yield better refractive outcomes: this is because treating the corneal edema first makes it easier to precisely identify the refractive target and the implants (including multifocal implants) that will best meet that target [49, 50]. Moreover, DMEK in phakic and young patients will help preserve the function of accommodation [51]. However, this approach must be balanced against the fact that all phacoemulsification techniques induce 8–15% ECL [52–

54], which could harm DMEK graft survival. Further research is needed to determine the benefits and disadvantages of this approach relative to triple-DMEK and pseudophakic-DMEK.

## Study limitations

This study is limited by the relatively small number of patients in each cohort, which may have limited our ability to detect significant differences for some variables. In addition, the study was retrospective. However, all data were recorded prospectively. Finally, the study was conducted in a single center. However, all surgeries were conducted by the same experienced surgeon, thus minimizing inter-surgeon differences.

## Supporting information

**S1 Table. Characteristics of the patient cohorts in the literature.**
(DOCX)

## Acknowledgments

We thank Dr. Yinka Zevering of SciMeditor Medical Writing Services for her assistance with this paper.

## Author Contributions

**Conceptualization:** Jean Marc Perone.

**Data curation:** Axelle Semler-Collery, Florian Bloch, George Hayek, Jean Marc Perone.

**Formal analysis:** Axelle Semler-Collery, Florian Bloch, George Hayek, Christophe Goetz.

**Funding acquisition:** Christophe Goetz, Jean Marc Perone.

**Investigation:** Axelle Semler-Collery, Jean Marc Perone.

**Methodology:** Christophe Goetz, Jean Marc Perone.

**Project administration:** Christophe Goetz, Jean Marc Perone.

**Resources:** Christophe Goetz, Jean Marc Perone.

**Software:** Christophe Goetz.

**Supervision:** Jean Marc Perone.

**Validation:** Jean Marc Perone.

**Visualization:** Axelle Semler-Collery.

**Writing – original draft:** Axelle Semler-Collery.

**Writing – review & editing:** Axelle Semler-Collery, Florian Bloch, George Hayek, Christophe Goetz, Jean Marc Perone.

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
