## [Decision Letter · Decision Letter 0]

25 Jan 2022

PONE-D-21-32369Comparison of triple-DMEK to pseudophakic-DMEK: a cohort study of 95 eyesPLOS ONE

Dear Dr. Perone,

Thank you for submitting your manuscript to PLOS ONE. After careful consideration, we feel that it has merit but does not fully meet PLOS ONE’s publication criteria as it currently stands. Therefore, we invite you to submit a revised version of the manuscript that addresses the points raised during the review process.

We look forward to receiving your revised manuscript.

Kind regards,

Timo Eppig

Academic Editor

PLOS ONE

Journal Requirements:

Reviewers' comments:

Reviewer's Responses to Questions

**Comments to the Author**

1. Is the manuscript technically sound, and do the data support the conclusions?

Reviewer #1: Partly

Reviewer #2: Yes

2. Has the statistical analysis been performed appropriately and rigorously? 

Reviewer #1: No

Reviewer #2: Yes

3. Have the authors made all data underlying the findings in their manuscript fully available?

Reviewer #1: No

Reviewer #2: Yes

4. Is the manuscript presented in an intelligible fashion and written in standard English?

Reviewer #1: Yes

Reviewer #2: Yes

5. Review Comments to the Author

Reviewer #1: The topic of this retrospective cohort study is of current interest.

Abstract

36-41 Separate preoperative and postoperative findings

38 (0.05 "vs." 0.05 logMAR)

Same facts should not be mentioned repeatedly in the abstract.

The following aspects should be rephrased:

- Similar complication rates <-- Rebubbling rates differ considerably (40% vs. 24%).

- “Better” spherical refraction in triple-DMEK <-- This results from different preconditions (Pre-DMEK Sph. Eq.: -0.13 vs. +0.23)

Preoperative examination

105 Why didn’t you use standard logMAR visual acuity charts (EDTRS), also for comparison with other studies

Surgical technique

154 Refractive target in triple-DMEK: "-"0.5? to -1.00

Statistical analyses

“All data were complete for each patient.” ?? “Fig. 1: 2 lost to follow up”

Mean+/-SD and Student’s t-test were used. How did you test for (and confirm) normal distribution?

Patient Selection

“Patients were excluded if the indication was not FECD and/or the patient had eye damage that could influence visual acuity…”

- 203 -212 Why did you include eyes with non-FECD and other pathologies (Fig.1: retinal detachment, AMD, vitreomacular traction,…) in the first place and excluded them afterwards?

Primary and secondary study outcomes

239 “triple-DMEK group tended to have a smaller residual hyperopia”

- But refractional target in triple-DMEK was -0.5 to -1.0 D and in the pseudophakic group preop. Sph. Eq. was already hyperopic (+0.23 D)

265 ”40% required one re-bubbling versus 24% for the pseudophakic-DMEK group (p=0.09)” - Please check significance.

Discussion

277-279 Rephrase statements regarding “the only” differences between both methods, hyperopia and rebubbling

318-319 “the greater ECL in triple DMEK may not have clinical consequences in the long term” – This sentence should be removed. See your findings: Triple-DMEK was associated with consistently lower ECD at 3, 6, 12 month and at 12 months, the mean ECL was 54% vs. 48% (236-238). The consequences of progredient cell loss (in total >1000 in the first 12 months) should not be underestimated.

330-331 Rebubbling rates (see above)

358-361 “Hyperopic differences” (see above)

374 Separate conclusion for study methods and discussion of a third approach

377 Complication rates (see above)

379-381 Repeated statement

Reviewer #2: Dear editor,

Dear authors,

thank you for inviting me to review this manuscript that analyzes outcomes in triple-DMEK and pseudophakic-DMEK based on 95 eyes.

Comments:

1 Line 56: “DMEK is the treatment of choice for FECD“. This does not apply in general: The latest 2019 Eye Banking Statistical Report of the United States of America indicates that DMEK has not yet surpassed DSEK (see: https://restoresight.org/wp-content/uploads/2020/04/2019-EBAA-Stat-Report-FINAL.pdf, page 7).

2 Line 63: “In 2016, 93% of all corneal transplants in the USA were conducted for FECD“. This is not true – then, there would be only 7% other corneal diseases requiring corneal transplantation in the United States for that year ... The 2016 Eye Banking Statistical Report of the United States of America reports that “Fuchs‘ dystrophy was the most common indication for keratoplasty again in 2016 (17,016, 23.3% (!)). (…) 93.1% of patients with Fuchs‘ dystrophy were treated with EK“. (see: http://restoresight.org/wp-content/uploads/2017/04/2016_Statistical_Report-Final-040717.pdf , page 10). The authors may would like to quote the original data from the Eye Banking Report which is freely available.

3 Lines 71: The reader would be interested in more details about these studies. How many patients were these studies based on ? How old were the patients ? Was the stage of FECD comparable in the different studies ?

4 Line 87: Typo clincaltrials.gov.

5 Line 106: I would suggest using “near visual acuity“ instead of “close visual acuity“.

6 Line 108: Which device was used for anterior segment imaging ?

7 Line 197: Were the groups examined for normal distribution before selecting the statistical test ?

8 Table 1: I do not understand the number of patients indicated under “female sex“ in the “Triple-DMEK“ group. The “Triple-DMEK“ group consisted of 34 patients. How can there be 37 patients with female sex in this group ? What do the percentages relate to ?

9 Line 329 and following: Rebubbling rates also depend on whether air or SF6-gas is used in DMEK procedure. Are the results based on air or gas tamponade ?

I hope my comments will help you to improve your manuscript.

6. PLOS authors have the option to publish the peer review history of their article (what does this mean?). If published, this will include your full peer review and any attached files.

Reviewer #1: No

Reviewer #2: No

---

## [Author Response · Author response to Decision Letter 0]

10 Mar 2022

Responses to reviewers

We would like to thank the reviewers very much for their thoughtful reviews of our manuscript and their helpful comments. We believe our paper is much improved after addressing these comments.

Reviewer #1: 

The topic of this retrospective cohort study is of current interest.

Reply: Thank you.

Abstract

36-41 Separate preoperative and postoperative findings

Reply: We separated the preoperative and postoperative findings as follows:

“They also did not differ in final BSCVA (both 0.03 logMAR), final endothelial-cell loss (54% vs. 48%), or astigmatism (-1.25 vs. -1 D). At 12 months, triple-DMEK associated with significantly smaller residual hyperopia (0.75 vs. 1 D; p=0.04) and spherical equivalence (0 vs. 0.5 D; p=0.02).” (lines 38–41)

38 (0.05 "vs." 0.05 logMAR)

Reply: This was changed to:

“They also did not differ in final BSCVA (both 0.03 logMAR)” (line 38)

Same facts should not be mentioned repeatedly in the abstract.

The following aspects should be rephrased:

- Similar complication rates <-- Rebubbling rates differ considerably (40% vs. 24%).

Reply: To address these points, we changed the last sentences as follows:

“In conclusion, while triple-DMEK and pseudophakic-DMEK achieved similar visual acuity improvement, triple-DMEK was superior in terms of final spherical refraction but also tended to have higher complication rates.” (lines 42–45)

- “Better” spherical refraction in triple-DMEK <-- This results from different preconditions (Pre-DMEK Sph. Eq.: -0.13 vs. +0.23)

Reply: The triple-DMEK and pseudophakic-DMEK patients did not differ in terms of baseline spherical equivalent (p=0.31; Table 1). 

Preoperative examination

105 Why didn’t you use standard logMAR visual acuity charts (EDTRS), also for comparison with other studies

Reply: It is standard practice in France to use the Monoyer scale. We convert the data into logMAR for statistical reasons. 

Surgical technique

154 Refractive target in triple-DMEK: "-"0.5? to -1.00

Reply: Yes, it was a typo, -0.5 is correct. This change has been made. (line 155)

Statistical analyses

“All data were complete for each patient.” ?? “Fig. 1: 2 lost to follow up”

Reply: The inclusion criteria included patients who were followed up for at least 12 months. To make this point clearer, we changed the eligibility criteria as follows:

“The patient cohort consisted of all consecutive adult (≥18 years) patients with FECD and BSCVA ≤0.3 LogMAR who underwent triple-DMEK or pseudophakic-DMEK in January 2015–January 2019 in the Department of Ophthalmology of the Metz-Thionville Regional Hospital Center (Metz, France) and who were followed up for at least 12 months. Patients were excluded if the patient was lost to 12-month follow-up and/or the patient had eye damage that could influence visual acuity recovery” (lines 95–100)

We also changed a sentence in the Statistics section as follows:

“The data were complete for all patients included in the analysis.” (line 194)

Mean+/-SD and Student’s t-test were used. How did you test for (and confirm) normal distribution?

Reply: Thank you very much for this point. We apologize: due to several people working on the first draft simultaneously, we accidentally omitted to actually conduct the distribution analyses. In fact, all variables except for ECD are not normally distributed. We have repeated all analyses according to their distribution. The new results are shown in Tables 1 and 3 and Figure 2 and the corresponding changes have been made in the manuscript. There were no major changes except that the difference between the triple-DMEK and pseudophakic-DMEK groups in terms of 12-month Sphere is now statistically significant (0.75 vs. 1 D, p=0.04) (in the original paper, it was 0.84 vs. 1.37 D, p=0.06). 

To address these changes, we altered some text in:

• The Abstract (lines 38–41)

• The Statistics section (lines 197–200)

• The Results (Tables 1 and 3, Figure 2, lines 221–222, 238–246)

• The Discussion (lines 282–284)

Patient Selection

“Patients were excluded if the indication was not FECD and/or the patient had eye damage that could influence visual acuity…”

- 203 -212 Why did you include eyes with non-FECD and other pathologies (Fig.1: retinal detachment, AMD, vitreomacular traction,…) in the first place and excluded them afterwards?

Reply: We deleted the non-FECD exclusion criterion. The eligibility criteria now read:

“The patient cohort consisted of all consecutive adult (≥18 years) patients with FECD and BSCVA ≤0.3 LogMAR who underwent triple-DMEK or pseudophakic-DMEK in January 2015–January 2019 in the Department of Ophthalmology of the Metz-Thionville Regional Hospital Center (Metz, France) and who were followed up for at least 12 months. Patients were excluded if the patient was lost to 12-month follow-up and/or the patient had eye damage that could influence visual acuity recovery” (lines 95–100)

Primary and secondary study outcomes

239 “triple-DMEK group tended to have a smaller residual hyperopia”

- But refractional target in triple-DMEK was -0.5 to -1.0 D and in the pseudophakic group preop. Sph. Eq. was already hyperopic (+0.23 D)

Reply: The difference between the triple-DMEK and pseudophakic-DMEK patients in terms of SE was not significant (p=0.31). The fact that pseudophakic-DMEK patients had worse SE than triple-DMEK patients after surgery (despite not differing significantly in baseline SE) is logical given that (i) DMEK is known to have a slight hypermetropizing effect and (ii) we can anticipate that in triple-DMEK and compensate for it during phacoemulsification, whereas we cannot compensate for it in patients who are already pseudophakic. This is the advantage of triple-DMEK over pseudophakic-DMEK. 

To address this point, we added a further clarifying point to the Discussion text:

“The better spherical equivalence after triple-DMEK reflects the fact that by conducting phacoemulsification at the same time as DMEK, we can preventively correct the slight hypermetropization that is observed after DMEK (whereas we cannot do that in pseudophakic DMEK).” (lines 371–374)

265 ”40% required one re-bubbling versus 24% for the pseudophakic-DMEK group (p=0.09)” - Please check significance.

Reply: The p value is correct. It is possible that with a larger sample size, this p value would have become significant. To address this, we added the sentence to the Rebubbling section in the Discussion:

“We observed that triple-DMEK tended to associate with greater rebubbling rates than pseudophakic-DMEK (40% vs. 24%; p=0.09). It is possible that with a larger sample size, this value would have become significant.” (lines 334–336)

Discussion

277-279 Rephrase statements regarding “the only” differences between both methods, hyperopia and rebubbling

Reply: The text was rephrased as follows:

“However, the triple-DMEK group had significantly smaller residual hyperopia (p=0.04) and spherical equivalent (p=0.02) and a tendency to more frequent rebubbling (p=0.09).” (lines 282–284)

318-319 “the greater ECL in triple DMEK may not have clinical consequences in the long term” – This sentence should be removed. See your findings: Triple-DMEK was associated with consistently lower ECD at 3, 6, 12 month and at 12 months, the mean ECL was 54% vs. 48% (236-238). The consequences of progredient cell loss (in total >1000 in the first 12 months) should not be underestimated.

Reply: Indeed, this endothelial cell loss can be harmful in the long term and should not be underestimated. The sentence was deleted. We also added the following sentence to the Conclusions:

“Evaluation of the long-term consequences of triple-DMEK may be needed to ensure its relative safety.” (lines 384–385)

330-331 Rebubbling rates (see above)

Reply: The p value is correct. It is possible that with a larger sample size, this p value would have become significant. To address this, we added the sentence to the Rebubbling section in the Discussion:

“We observed that triple-DMEK tended to associate with greater rebubbling rates than pseudophakic-DMEK (40% vs. 24%; p=0.09). It is possible that with a larger sample size, this value would have become significant.” (lines 334–336)

358-361 “Hyperopic differences” (see above)

Reply: The difference between the triple-DMEK and pseudophakic-DMEK patients in terms of SE was not significant (p=0.31). The fact that pseudophakic-DMEK patients had worse SE than triple-DMEK patients after surgery (despite not differing significantly in baseline SE) is logical given that (i) DMEK is known to have a slight hypermetropizing effect and (ii) we can anticipate that in triple-DMEK and compensate for it during phacoemulsification, whereas we cannot compensate for it in patients who are already pseudophakic. This is the advantage of triple-DMEK over pseudophakic-DMEK. 

To address this point, we added a further clarifying point to the Discussion text:

“The better spherical equivalence after triple-DMEK reflects the fact that by conducting phacoemulsification at the same time as DMEK, we can preventively correct the slight hypermetropization that is observed after DMEK (whereas we cannot do that in pseudophakic DMEK).” (lines 371–374)

374 Separate conclusion for study methods and discussion of a third approach

Reply: To address this point, we created two sections called “Overall study conclusions” and “Discussion of a recent third approach (DMEK in phakic patients)” (lines 380 and 389)

377 Complication rates (see above)

Reply: We deleted “and similarly low complication rates”.

379-381 Repeated statement

Reply: We changed the text as follows:

“In our study, triple-DMEK and pseudophakic-DMEK both associated with marked improvements in postoperative visual acuity. Triple-DMEK associated with significantly better final spherical refractive outcomes, although it also tended to induce more rebubbling and ECL. Evaluation of the long-term consequences of triple-DMEK may be needed to ensure its relative safety. Nonetheless, our findings are important because combining phacoemulsification with DMEK prophylactically prevents progression of corneal impairment and eliminates a second surgery, thus reducing surgical risk [32]. It is also more cost-effective for the patient [32,34,35].” (lines 381–387)

Reviewer #2: 

Dear editor,

Dear authors,

thank you for inviting me to review this manuscript that analyzes outcomes in triple-DMEK and pseudophakic-DMEK based on 95 eyes.

Comments:

1 Line 56: “DMEK is the treatment of choice for FECD“. This does not apply in general: The latest 2019 Eye Banking Statistical Report of the United States of America indicates that DMEK has not yet surpassed DSEK (see: https://restoresight.org/wp-content/uploads/2020/04/2019-EBAA-Stat-Report-FINAL.pdf, page 7).

Reply: We agree. Because DMEK is more difficult to perform, DSAEK is most often used globally to treat FECD. We qualified the sentence as follows:

“Although DMEK is more difficult to perform than DSAEK, it is often the treatment of choice for FECD [1,11,12] because it associates with better recovery, postoperative best spectacle-corrected visual acuity (BSCVA) [12–14], contrast [15,16], immune rejection [17,18], patient satisfaction [12,19,20], and final posterior residual corneal higher-order aberrations [21].” (lines 56–60)

2 Line 63: “In 2016, 93% of all corneal transplants in the USA were conducted for FECD“. This is not true – then, there would be only 7% other corneal diseases requiring corneal transplantation in the United States for that year ... The 2016 Eye Banking Statistical Report of the United States of America reports that “Fuchs‘ dystrophy was the most common indication for keratoplasty again in 2016 (17,016, 23.3% (!)). (…) 93.1% of patients with Fuchs‘ dystrophy were treated with EK“. (see: http://restoresight.org/wp-content/uploads/2017/04/2016_Statistical_Report-Final-040717.pdf , page 10). The authors may would like to quote the original data from the Eye Banking Report which is freely available.

Reply: We apologize, it was a typo. The text was changed as follows:

“FECD is the most common corneal dystrophy: in 2016, it accounted for 23% of all corneal transplants in the USA” (line 64)

3 Lines 71: The reader would be interested in more details about these studies. How many patients were these studies based on ? How old were the patients ? Was the stage of FECD comparable in the different studies ?

Reply: We showed these details in a new table designated as Supplementary Table S1.

4 Line 87: Typo clincaltrials.gov.

Reply: The typo was fixed. (line 88)

5 Line 106: I would suggest using “near visual acuity“ instead of “close visual acuity“.

Reply: This change was made. (line 107)

6 Line 108: Which device was used for anterior segment imaging ?

Reply: The RS-3000 OCT RetinaScan advance NIDEK, Co., Ltd. This was added to the manuscript. (lines 108–109).

7 Line 197: Were the groups examined for normal distribution before selecting the statistical test ?

Reply: Thank you very much for this point. We apologize: due to several people working on the first draft simultaneously, we accidentally omitted to actually conduct the distribution analyses. In fact, all variables except for ECD are not normally distributed. We have repeated all analyses according to their distribution. The new results are shown in Tables 1 and 3 and Figure 2 and the corresponding changes have been made in the manuscript. There were no major changes except that the difference between the triple-DMEK and pseudophakic-DMEK groups in terms of 12-month Sphere is now statistically significant (0.75 vs. 1 D, p=0.04) (in the original paper, it was 0.84 vs. 1.37 D, p=0.06). 

To address these changes, we altered some text in:

• The Abstract (lines 38–41)

• The Statistics section (lines 197–200)

• The Results (Tables 1 and 3, Figure 2, lines 221–222, 238–246)

• The Discussion (lines 282–284)

8 Table 1: I do not understand the number of patients indicated under “female sex“ in the “Triple-DMEK“ group. The “Triple-DMEK“ group consisted of 34 patients. How can there be 37 patients with female sex in this group ? What do the percentages relate to ?

Reply: We apologize, the numbers were accidentally reversed. Women accounted for 27 of the 34 triple DMEK patients (79%) and 37 of the 43 pseudophakic-DMEK patients (86%). Table 1 has been altered to reflect this.

9 Line 329 and following: Rebubbling rates also depend on whether air or SF6-gas is used in DMEK procedure. Are the results based on air or gas tamponade ?

Reply: We used air tamponade for all cases in this series (we have since switched to SF6). This point was added to the Methods:

“If central graft detachment was observed or more than a third of the graft had detached, rebubbling with an air tamponade was performed” (lines 172–175)

I hope my comments will help you to improve your manuscript.

Reply: Thank you very much for your thorough review of our manuscript, your comments have improved it significantly.

---

## [Decision Letter · Decision Letter 1]

6 Apr 2022

PONE-D-21-32369R1Comparison of triple-DMEK to pseudophakic-DMEK: a cohort study of 95 eyesPLOS ONE

Dear Dr. Perone,

Thank you for submitting your manuscript to PLOS ONE. After careful consideration, we feel that it has merit but does not fully meet PLOS ONE’s publication criteria as it currently stands. Therefore, we invite you to submit a revised version of the manuscript that addresses the points raised during the review process. Please check for the remaining errors identified by the reviewer.

We look forward to receiving your revised manuscript.

Kind regards,

Timo Eppig

Academic Editor

PLOS ONE

Journal Requirements:

Reviewers' comments:

Reviewer's Responses to Questions

**Comments to the Author**

1. If the authors have adequately addressed your comments raised in a previous round of review and you feel that this manuscript is now acceptable for publication, you may indicate that here to bypass the “Comments to the Author” section, enter your conflict of interest statement in the “Confidential to Editor” section, and submit your "Accept" recommendation.

Reviewer #2: All comments have been addressed

2. Is the manuscript technically sound, and do the data support the conclusions?

Reviewer #2: Yes

3. Has the statistical analysis been performed appropriately and rigorously? 

Reviewer #2: Yes

4. Have the authors made all data underlying the findings in their manuscript fully available?

Reviewer #2: Yes

5. Is the manuscript presented in an intelligible fashion and written in standard English?

Reviewer #2: Yes

6. Review Comments to the Author

Reviewer #2: The authors have revised their manuscript according to the reviewers' suggestions.

However, a minor issue needs to be resolved:

The data shown in Table 1 does not correspond to the data about their own study (!) as shown in Supplemental Table S1.

Mean age of patients in the study is stated to be 70 (Triple DMEK) and 72 (Pseudophakic-DMEK) in table 1.

Supplemental Table S1 states "72 vs. 69".

Same for the number of patients (55(43) vs. 40(34) in table 1 and 55(43) vs. 34(40) in supplemental table S1).

Same for female sex (79% and 86% in table 1 versus 68% and 67% in supplemental table S1).

Same for BSCVA (0.50 in table 1 and 0.63/0.59 in supplemental table S1).

Actually, proofreading and noticing of such errors should be done by the authors themselves !

7. PLOS authors have the option to publish the peer review history of their article (what does this mean?). If published, this will include your full peer review and any attached files.

Reviewer #2: No

---

## [Author Response · Author response to Decision Letter 1]

8 Apr 2022

Reply to Reviewer

Reviewer #2: The authors have revised their manuscript according to the reviewers' suggestions.

However, a minor issue needs to be resolved:

The data shown in Table 1 does not correspond to the data about their own study (!) as shown in Supplemental Table S1.

Mean age of patients in the study is stated to be 70 (Triple DMEK) and 72 (Pseudophakic-DMEK) in table 1.

Supplemental Table S1 states "72 vs. 69".

Same for the number of patients (55(43) vs. 40(34) in table 1 and 55(43) vs. 34(40) in supplemental table S1).

Same for female sex (79% and 86% in table 1 versus 68% and 67% in supplemental table S1).

Same for BSCVA (0.50 in table 1 and 0.63/0.59 in supplemental table S1).

Actually, proofreading and noticing of such errors should be done by the authors themselves !

Reply: We sincerely apologize for this error and are grateful that you picked it up. The previous review identified errors in Table 1 that we corrected but then we overlooked correcting the data in Supplementary Table S1 as well. These corrections have been made.

---

## [Editor Report · Decision Letter 2]

20 Apr 2022

Comparison of triple-DMEK to pseudophakic-DMEK: a cohort study of 95 eyes

PONE-D-21-32369R2

Dear Dr. Perone,

We’re pleased to inform you that your manuscript has been judged scientifically suitable for publication and will be formally accepted for publication once it meets all outstanding technical requirements.

Kind regards,

Timo Eppig

Academic Editor

PLOS ONE
---

## [Editor Report · Acceptance letter]

26 Apr 2022

PONE-D-21-32369R2 

Comparison of triple-DMEK to pseudophakic-DMEK: a cohort study of 95 eyes 

Dear Dr. Perone:

I'm pleased to inform you that your manuscript has been deemed suitable for publication in PLOS ONE. Congratulations! Your manuscript is now with our production department. 

Kind regards, 

on behalf of

Dr. Timo Eppig 

Academic Editor

PLOS ONE